# Farmers’ Knowledge, Attitude, and Adoption of Smart Agriculture Technology in Taiwan

**DOI:** 10.3390/ijerph17197236

**Published:** 2020-10-03

**Authors:** Jui-Hsiung Chuang, Jiun-Hao Wang, Yu-Chang Liou

**Affiliations:** Department of Bio-Industry Communication and Development, National Taiwan University, Taipei City 10617, Taiwan; d06630001@ntu.edu.tw (J.-H.C.); d03630003@ntu.edu.tw (Y.-C.L.)

**Keywords:** smart agriculture, agriculture 4.0, innovation adoption, digital technology, Taiwan

## Abstract

Climate change and food security are critical topics in sustainable agricultural development. The climate-smart agriculture initiative proposed by the Food and Agriculture Organization of the United Nations has attracted international attention. Smart agriculture (SA) has since been recognized as an influential trend contributing to agricultural development. Therefore, encouraging farmers to adopt digital technologies and mobile devices in farming practices has become a policy priority worldwide. However, the literature on the psychological factors driving farmers’ intentions to adopt SA technologies remains limited. This study investigated how farmers’ knowledge and attitudes regarding SA affect their adoption of smart technologies in Taiwan. A total of 321 farmers participated in a survey in 2017 and 2018, and the data were used to construct an ordinary least squares regression model of SA adoption. This study provides a preliminary understanding of the relationship between psychological factors and innovation adoption of SA technologies in a small-scale farming economic context. The findings suggest that policymakers and research and development institutes should concentrate on improving market access to established and critical SA technologies.

## 1. Introduction

Climate change and food security have become critical topics in sustainable agricultural development. The Food and Agriculture Organization of the United Nations proposed the climate-smart agriculture (CSA) concept, which has attracted international attention for its innovative use of technology in addressing agricultural challenges [1,2]. The objectives of CSA are threefold: sustainably increase food productivity, increase the adaptive capacity of farming systems, and increase climate change mitigation where possible [3,4,5]. Smart agriculture (SA) emphasizes the roles and applications of innovative technology in agricultural practices.

The SA strategy focuses on the use of digital technology to create precision farming solutions, especially when combined with the application of information and communication technologies and other new interconnected equipment and techniques. The internet of things (IoT), drones, robots, big data, cloud computing, and artificial intelligence are all new resources that are expected to be applied to novel farming practices [6]. The integration of precision farming systems and digital technology has become the most prevalent trend in agricultural development, contributing to fewer inputs, higher yields, and less damage in agricultural production. Digitized agriculture has become a mainstream trend in numerous countries [7,8,9,10].

Several similar but inconclusive concepts were used in different studies, such as agriculture 4.0, precision agriculture, smart farming, digital agriculture, virtual agriculture, big data in agriculture, IoT in agriculture, and interconnected agriculture [11,12]. These emerging concepts have slight differences in terms of the emphasis on specific technological applications. Most of these concepts share common traits and values in incorporating new and intelligent technologies into farming practices and introducing resource use efficiency approaches that minimize production costs, reduce farming risks, and increase productivity [13]. The inventory of the European smart agricultural knowledge and innovation systems (Smart-AKIS) program indicates that SA is mainly related to three interconnectable new technology categories: farm management information systems, precision farming, and agricultural automation. For instance, smartphone application software has been extensively used for remote monitoring and controlling of farming equipment. A similar phenomenon is observed in plant factories, which have employed IoT, big data, sensing and monitoring techniques, and automatic environmental control systems [7]. Therefore, agribusiness and small-scale farmers can benefit from the application of new technologies.

Taiwan’s agricultural sector is characterized by small-scale holdings and has been identified as a global disaster hotspot (e.g., typhoons and floods) [14]. The Agriculture 4.0 Project was launched by the Council of Agriculture (COA) of Taiwan in 2017 in compliance with Industry 4.0 development and climate change risks. In the pilot project, an attempt was made to introduce advanced technologies, such as intelligent devices, sensing techniques, robots, IoT, and big data analysis to improve agricultural productivity. The government of Taiwan has invested approximately TWD 4.5 billion in upgrading agricultural technologies. The project was renamed the Smart Agriculture Project in 2018. The SA Project aimed to overcome the restrictions of natural resources and shortages in human labor resources by facilitating the intelligent production and digital marketing of agricultural businesses [15]. The principal strategies of the SA Project were threefold. First, the COA selected ten pilot agribusinesses as the prioritized targets for the first stage of SA promotion. The agribusinesses targeted were those for moth orchids, seedlings, mushrooms, rice, agricultural facilities, aquaculture, poultry, traceable agricultural products, dairy, and offshore fisheries. Second, the agricultural research and development (R&D) institute employed cross-domain technological innovations to create digital agri-services, value chains, and communication models between producers and consumers, such as IoT-based environmental control modules, labor-saving carrying equipment, and marketing management information platforms. Third, the next generation of farmers should satisfy requirements for the use of smart agricultural development because trained farmers are the foundation of SA development [16,17].

Human resource development is a key factor in developing SA; thus, encouraging farmers and agribusinesses to adopt innovative digital technologies and intelligent mobile devices in their farming practices is becoming a policy priority in Taiwan. Therefore, the COA and National Taiwan University collaborated to create and design a series of SA training programs to develop human resources in smart agriculture. The educational objectives of the SA training program were to confer trainees with a positive attitude and practical competences, and to enhance their SA-related knowledge. Four types of training courses were offered, comprising of indoor lectures providing general SA education, on-site visits and training, international visits and exchanges, and individual tailor-made technical assistance provided by SA service teams for each pilot SA industry [17]. However, the literature on the psychological factors and individual characteristics that drive farmers’ intentions to adopt SA technologies remains limited. Therefore, this study investigated the associations among SA-related knowledge, attitudes, and adoption behaviors. Moreover, we assessed the effect of farmers’ knowledge and attitudes regarding smart agriculture on their adoption of SA technologies.

The following research problems regarding SA knowledge, attitude, and adoption were addressed. What types of SA technology are crucial for farming practices and are better understood by farmers? What are the driving factors in SA adoption behaviors? To what extent do sociodemographic variables, knowledge, and attitude affect the adoption of SA technologies?

Studies have focused on the effects of psychological factors on individual behaviors, such as social learning theory, the theory of reasoned action, and the theory of planned behavior [18,19]. Few studies on agriculture have identified associations between farming practices, attitudes, and other psychological determinants [13,20]. Furthermore, the theory of planned behavior, an extension of the theory of reasoned action, has been extensively applied and tested in various fields [21]. The theory of planned behavior identifies hierarchical relations between various beliefs and attitudes affecting behavior. The educational goals of agricultural training programs are multifaceted; such programs are expected to improve the target group’s knowledge level and change their attitudes and adoption behaviors [13,22]. This study employed a comprehensive knowledge–attitude–practice (KAP) model based on the previous literature, to further investigate the relationships in the KAP model of participants in the SA training course. Based on the KAP model, we hypothesized that SA knowledge and perceived importance were positively correlated and that both the SA knowledge and importance perception had a positive effect on the adoption of smart agriculture technologies.

## 2. Data and Measures

### 2.1. Data and Samples

The data used in this study were drawn from a survey of trainees of the SA training program conducted in the summers of 2017 and 2018. The training program was sponsored by the COA in Taiwan. All participants were asked to complete the survey through face-to-face interviews as a reference for training course planning. The sample characteristics are presented in Table 1. Among the 321 respondents, 79.1% were men, the average age was 42.61 years old, and 15.3% and 58.6% graduated from senior high school or below and college, respectively. Furthermore, 12.8%, 22.7%, and 64.5% of respondents were principal operators, hired staff of agribusinesses, and self-employed farmers, respectively. The average farm size was 3.9 hectares, the annual turnover was <TWD 0.2 million, TWD 0.2–1 million, TWD 1–5 million, and >TWD 5 million in 25.9%, 28.3%, 26.8%, and 19.0% of cases, respectively.

The main purpose of this study was to explore the knowledge, attitudes, and practices of SA trainees. The questionnaire design was developed based on previous research, as discussed in the literature review section. The dependent variable was SA adoption, which refers to the self-reported adoption level of SA technology in farming practices. To ensure the dependent variable measurements were valid and reliable, photographs of common SA technologies were presented in the interview. Respondents were thus provided with a framework from which to rate the adoption level of the SA technology from 0 (the lowest adoption) to 100 (the highest adoption).

The principal independent variables in this study were the levels of knowledge and the perceived importance of each type of SA technology. The Smart-AKIS inventory of SA technologies [7] indicates that abundant farming facilities and equipment for smart technology have been developed in Taiwan, including IoT devices, wireless sensors, monitoring equipment with automated climate data acquisition (climate sensing and monitoring), biological image detection and recognition equipment (image recognition), cloud and big data analysis services (big data), mobile phone apps for farm management, robotic farming machines, spraying and aerial photography drones, and automatic environmental control systems.

Respondent knowledge of eight types of SA technologies was assessed. Reference answers were as follows: “Never heard of it (=1)”, “Heard of it but do not know much about it (=2)”, “Have a general understanding of it (=3)”, “Understand it well and can explain it to others (=4)”. The same scale was used to measure respondents’ perceptions of the importance of adopting SA technology. The level of perceived SA importance was measured by asking “To what extent do you think that SA technology is important for improving the management of your farm?”

All responses were scored on a 4-point Likert scale, with a larger score indicating a higher degree of knowledge and perceived importance (1 = not important at all to 4 = very important) of SA technology. The sociodemographic variables measured were gender, age (in years), education level, and farmer type. Moreover, the farm features recorded were farm size (in hectares) and average annual turnover.

### 2.2. Statistical Analysis

The primary purpose of this study was to investigate the effect of psychological factors on the adoption behavior of SA technology, after individual sociodemographic and farm characteristics were controlled for. The empirical analysis was performed in two steps. First, this study explored the associations between farmers’ knowledge, perceived importance, sociodemographic characteristics, and adoption of SA technology behavior. The second stage of the analysis focused on investigating the effects of exogenous determinants on farmers’ adoption of SA technology behaviors. The following equation is the SA adoption function, which was used to calculate the relationship between the adoption level and SA knowledge, SA perceived importance, and sociodemographic and farm characteristics. The corresponding ordinary least squares (OLS) regression equation is specified as follows:(1)$$SA\_Adoption_{i}=\alpha_{0}+\beta^{\prime }Knowl_{i}+\gamma^{\prime }Imp_{i}+\lambda^{\prime }Z_{i}+\nu^{\prime }R_{i}+\varepsilon_{i}$$
where *SA_Adoption_i_* is individual *i*’s self-reported score for SA adoption behavior; *Knowl_i_* is the SA knowledge score of individual *i*; *Imp_i_* is the SA perceived importance score for individual *i*; *Z_i_* is a set of sociodemographic characteristics; and *R_i_* represents a set of farming features. The random error term $\varepsilon_{i}$ is assumed to be normally distributed, and $\alpha_{0},\;\beta^{’},\;\gamma^{’},\;\lambda^{\prime },\;\nu^{\prime }$ are the coefficients to be estimated. The $\;\beta^{’}\;\mathrm{and}\;\gamma^{’}$ coefficients are of particular interest because they represent the effect of SA knowledge and perceived importance on SA adoption. All analyses were performed using the SPSS software version 22.0 (SPSS Inc., Chicago, IL, USA).

## 3. Results and Discussion

### 3.1. Association between SA Knowledge, Importance and Adoption

The descriptive statistics of the knowledge, importance, and adoption level of SA technologies are illustrated in Table 2. The average self-reported score of SA adoption was 40.22, which corresponds to farming mechanization using a combustion engine or electricity. The mean total SA importance score was 25.87. Among all individual SA items, the SA technologies rated most important were, in order, automatic environmental control systems (3.24), farm management apps (3.35), and cloud and big data analysis services (3.33). These results indicate that automatic environmental control systems and farm management apps were the most familiar technologies to respondents and were perceived as being the most important smart technologies. The mean total knowledge score was 22.45. The three most well-known new SA technologies were, in order, automatic environmental control systems (3.04), spraying and aerial photography drones (2.93), and farm management apps (2.90).

The correlations between different indicators of SA knowledge, attitudes, and practices were investigated. Significant and positive associations were observed between all the KAP indicators. Table 3 and Table 4 present the means, standard deviations (SDs), and correlation coefficients between individual SA technologies. The results of the correlation between SA knowledge and adoption behavior are presented in Table 3. The individual coefficients range from 0.582 to 0.738. All the SA knowledge indicators were significantly and positively correlated with each other, as expected. The knowledge level of individual SA technologies was also significantly correlated with SA adoption. The three highest correlation coefficients were, in order, farm management apps (*r* = 0.306), cloud and big data analysis services (*r* = 0.296), and biological image detection and recognition techniques (*r* = 0.286). The results of the correlation matrix support the hypothesis of a positive relationship between SA knowledge and SA adoption behavior, which is consistent with the findings of studies on innovation adoption [2].

Similar results were revealed in the correlation matrix of SA importance and adoption behavior. Table 4 presents the correlation coefficients among the SA importance variables, which ranged from 0.319 to 0.667. All the SA importance indicators were positively and significantly correlated with each other. Most SA importance variables were significantly correlated with SA adoption. However, the correlation coefficients of the SA importance variables and SA adoption were lower than the coefficients of SA knowledge and SA adoption. The highest coefficient was 0.266 (automatic environmental control system). Furthermore, our results did not reveal a significant relationship between SA adoption and the perceived importance of image recognition techniques and drone technology. Because some correlations were nonsignificant, the SA importance hypotheses of SA adoption were only partially supported.

### 3.2. Effects of SA Knowledge and Importance on SA Adoption

Estimated results for the OLS multiple regression model of SA adoption are reported in Table 5. The adjusted R-square statistic indicated that 25.3% of the variation in SA adoption is explained by this regression model. Moreover, the F-value (10.02) of overall significance was below the significance level of 0.001. These results indicate that knowledge of SA, perceived importance of SA technologies, sociodemographic characteristics, and farm characteristics have a significant influence on SA adoption.

As illustrated in Table 5, SA knowledge and perceived importance were determined to be positively related to SA adoption. For example, a 1% increase in SA knowledge level was determined to increase the SA adoption score by 0.932% among respondents. Similarly, a 1% increase in SA importance level was determined to increase the SA adoption level by 0.811%. Our findings indicated that participants in the SA training program with higher levels of SA knowledge and perceived importance would adopt more innovative technologies in their farming practices. These findings are consistent with previous studies [23] that reported that knowledge and perceptions of SA technologies are critical determinants of innovation adoption behavior.

The relationships between other determinants and SA adoption behavior were also briefly discussed. The farm characteristics significantly affected the adoption level of SA technologies. For example, farmers working in the agribusiness sector exhibited higher adoption levels of SA technologies than self-employed farmers, as was expected [24]. Furthermore, the size and annual turnover of the farm were positively associated with SA adoption. These findings indicate that larger farm size or volume of business may enhance farmers’ investments in SA technologies. This finding accords with those of previous studies [13,25,26] that reported a positive relationship between farm size or revenue and innovation adoption. Most sociodemographic characteristics were not significantly associated with SA adoption, except age, which was positively correlated with SA adoption.

## 4. Conclusions

This study investigated SA-related knowledge, attitude, and adoption among farmers in Taiwan. The sociodemographic characteristics of the respondents were collected, and the effects on the adoption of SA technologies were determined. Survey data from 321 farmers who participated in the SA training program were collected. The results revealed significant and positive correlations between SA knowledge, perceived importance, and adoption behavior. Of the eight SA technologies, the automatic environmental control systems were the most well-understood and were perceived as being the most important, whereas biological image detection and recognition techniques were ranked as the least understood. SA knowledge and perceived importance significantly affected the adoption of SA technologies. Therefore, lower adoption levels of SA technologies may be attributed to inadequate information, missing knowledge, lack of awareness of the technologies, and lack of perceived practical value. We thus recommend the intensification of R&D and SA technologies, such as IoT and big data analysis, to satisfy farmer requirements under current farming conditions and management.

These findings provide policymakers and agricultural educators with important insights that can be used to more accurately target interventions that promote or facilitate the adoption of SA technologies. Furthermore, these findings suggest that agricultural R&D institutes should concentrate on improving market access for established and valuable SA technologies. Additionally, providing systematic training courses related to the applications of IoT and big data in agriculture may enable farmers to engage more effectively in SA practices. However, the research limitations should be considered when these findings are being interpreted. First, KAP strengthens the theoretical foundation of this study, however, it also limits its depth. Agriculture policy, organizational support, computer efficacy, perceived effectiveness, perceived usefulness, and trust toward SA ventures should be considered in future models. Second, in methodologies used to analyze the collected data, the mediating role of attitude on the relationship between knowledge and practice could have been investigated. Furthermore, the moderating effects of gender and prior experience could have also been examined.

## Figures and Tables

**Table 1 ijerph-17-07236-t001:** Descriptive statistics of sample characteristics (*n* = 321).

Variables		Frequency (Mean)	%	SD ^a^
Gender	Male	254	79.1	
	Female	67	20.9	
Age (years) ^b^		42.61		11.22
Edu level	Senior high or below	49	15.3	
	College/University	188	58.6	
	Graduated or above	84	26.2	
Farmer type	Owner or operator of Agribusiness	41	12.8	
	Hired staffs in Agribusiness	73	22.7	
	Self-employed	207	64.5	
Farm size (hectare) ^b^		3.92		13.57
Annual turnover (TWD)	0.2 million or below	83	25.9	
	0.2–1 million	91	28.3	
	1–5 million	86	26.8	
	5 million or above	61	19.0	

Note: ^a^ SD, standard deviation. ^b^ Age and farm size are presented as means and SDs.

**Table 2 ijerph-17-07236-t002:** Descriptive statistics of smart agriculture (SA) knowledge, importance and adoption (*n* = 321).

SA Technology	SA Importance			SA Knowledge		
	Mean	SD ^a^	Rank	Mean	SD ^a^	Rank
Total adoption score	40.22	20.82	-	-	-	-
Automatic control system	3.24	0.74	1	3.04	0.81	1
Apps	3.35	0.57	2	2.90	0.96	3
Big data	3.33	0.59	3	2.68	0.94	7
IoT	3.27	0.52	4	2.75	0.81	5
Image recognition	3.23	0.58	5	2.59	0.97	8
Sensing and monitoring	3.22	0.57	6	2.71	0.93	6
Robotic	3.12	0.63	7	2.85	0.83	4
Drones	3.10	0.65	8	2.93	0.84	2

Note: ^a^ SD, standard deviation.

**Table 3 ijerph-17-07236-t003:** Correlation matrix of the SA knowledge and adoption (*n* = 321).

SA Knowledge	1	2	3	4	5	6	7	8	9
1. IoT	1								
2. Climate sensing and monitoring	0.644 **	1							
3. Image recognition	0.590 **	0.722 **	1						
4. Big data	0.666 **	0.712 **	0.762 **	1					
5. Apps	0.638 **	0.657 **	0.691 **	0.717 **	1				
6. Robotic	0.583 **	0.610 **	0.582 **	0.632 **	0.610 **	1			
7. Drones	0.600 **	0.568 **	0.632 **	0.649 **	0.667 **	0.662 **	1		
8. Automatic system	0.597 **	0.602 **	0.638 **	0.695 **	0.645 **	0.645 **	0.738 **	1	
9. SA adoption score	0.251 **	0.219 **	0.286 **	0.296 **	0.306 **	0.249 **	0.224 **	0.270 **	1

Note: ** denotes significant differences at a *p* value < 0.01.

**Table 4 ijerph-17-07236-t004:** Correlation matrix of the SA importance and adoption (*n* = 321).

SA Importance	1	2	3	4	5	6	7	8	9
1. IoT	1								
2. Climate sensing and monitoring	0.458 **	1							
3. Image recognition	0.320 **	0.571 **	1						
4. Big data	0.537 **	0.509 **	0.505 **	1					
5. Apps	0.442 **	0.589 **	0.556 **	0.667 **	1				
6. Robotic	0.367 **	0.385 **	0.422 **	0.436 **	0.470 **	1			
7. Drones	0.319 **	0.344 **	0.488 **	0.418 **	0.402 **	0.530 **	1		
8. Automatic system	0.481 **	0.383 **	0.375 **	0.524 **	0.426 **	0.355 **	0.407 **	1	
9. SA adoption score	0.129 *	0.166 **	0.016	0.132 *	0.184 **	0.148 **	0.106	0.266 **	1

Note: ** denotes significant differences at a *p* value of <0.01; * denotes significant differences at a *p* value of < 0.05.

**Table 5 ijerph-17-07236-t005:** Estimation results of the ordinary least squares (OLS) regression (Dependent variable: SA adoption, *n* = 321).

Variable	Coefficient	s.e.	t-Value
Total_Knowledge	0.93 **	1.50	4.97
Total_Importance	0.81 *	2.54	2.56
Socio-demographic characteristics			
Male	3.66 *	2.52	1.45
Age	0.23	0.10	2.42
University	2.65	2.95	0.90
Graduated or above	−0.26	3.35	−0.08
Farming features			
Operator	6.75 *	3.24	2.08
Hired staffs	8.52 **	2.56	3.33
Farm size (ha)	0.15 *	0.08	1.99
Turnover_0.2–1 million	8.83 **	2.77	3.19
Turnover_1–5 million	15.75 **	2.87	5.48
Turnover_5 million and above	17.32 **	3.15	5.491
Intercept	−24.43	10.31	

Note: s.e. stands for standard error. The reference group for educational level is “Senior high or below”; farmer type is “Self-employed farmer”; annual turnover is <TWD 0.2 million. ** *p* < 0.01; * *p* < 0.05.

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
