# Peer review of "Farmers’ Knowledge, Attitude, and Adoption of Smart Agriculture Technology in Taiwan"

_ijerph, 2020, doi:10.3390/ijerph17197236_

Round 1

Reviewer 1 Report

The article submitted for review is an interesting contribution to further research. The theoretical description of the work is an introductory introduction to a very wide issue. There is no reference to similar research conducted in other countries, which would give the possibility of a broader view of the topic under study. The research sample itself is relatively small and concerns only one country, which means that the article has no chance of attracting potential readers.
The research methodology leaves wele to be desired ... I believe that the article would be more suitable for a national, trade magazine.

Author Response

Reviewer 1:

  1. The article submitted for review is an interesting contribution to further research. The theoretical description of the work is an introductory introduction to a very wide issue. There is no reference to similar research conducted in other countries, which would give the possibility of a broader view of the topic under study. The research sample itself is relatively small and concerns only one country, which means that the article has no chance of attracting potential readers.

Our responses (#1.1):

  • We greatly appreciate that you recognized our efforts in conceptualizing this study. We have tried our best to make a balance in responding to all the issues identified by the reviewers. All revised texts are highlighted in red.
  • We have added several references of similar research conducted in other countries. Please refer to page 1 of the revised document.
  • Research samples in farmer survey studies usually range from 100 to 500. For example, the sample sizes in the studies by Pivoto et al. (2019) in Brazil, Pagliacci et al. (2020) in Italy, and Chuang et al. (2020) in Taiwan were 119, 463, and 241, respectively. The sample of the present study is 321, which may not be criticized as “relatively small.”
  • International Journal of Environmental Research and Public Health (IJERPH) “provides comprehensive and unique information with a worldwide readership”. Most articles published in the IJERPH focus on comprehensive and unique information in a specific nation or area rather than cross-national comparisons.

  1. The research methodology leaves were to be desired ... I believe that the article would be more suitable for a national, trade magazine.

Our responses (#1.2):

We cherish this comment and put it into the research limitation. Please refer to page 8 of the revised document.

=====================

In sum, we have tried our best to respond to all the issues identified by the reviewers. Besides, the revised document was edited by a professional editing company. We attached the proof issued by that company, as follows. As a result, we believe that the manuscript has been strengthened and thus we are now resubmitting the manuscript for your further review. Your consideration and feedback are highly appreciated, and we look forward to hearing from you in the near future.

Reviewer 2 Report

The research presented in this paper is interesting . Nevertheless, I think that some things should be solved. Most of the English issues detected were marked and there possible correction is in the text.

There are 9 sentences, marked with yellow that I think should be re-written or clarified. These aspects are more clear in a separate file.

1. I think the sentence should be re-written as:

Due to integration of precision farming system and digital technology, SA has been recognized as the most prevalent trend in agricultural development, contributing to fewer inputs, higher yields and less damage of agricultural production.

2. I think the sentence should be re-written as:

Despite the similarities there are slight differences between those emerging concepts, namely the emphasis on specific technological applications.

3. The research problems of this study will address the following questions regarding knowledge, attitude and adoption in relation to SA technology:

  • What types of SA technology are important for farming practices and better understood by farmers?
  • What are the driving factors for SA adoption behaviors?
  • To what extent the socio-demographic variables, knowledge and attitude may be associated with the adoption of SA technologies.

4. There is something missing in this sentence. What is due to the educational goals of an agricultural training program? It seems we are missing the end of the sentence.

5. I don’t think we should refer to farmers/trainees as research subjects. So I think this sentence should be re-written as:

The respondents were trainees of the SA training program, sponsored by the COA in Taiwan.

6. As it is written, it seems that the age and educational level only refer to the males. As I think this is not true, the sentence should be re-written as:

Among 321 respondents, of which 79.1% were male, the average age was 42.61 years old and 15.3% and 58.6% of them were, respectively, graduated from senior high school or below and college.

7. I don’t know exactly what do you mean with this sentence. If I’m not mistaken, you want to say that you used the same scale and not the same indicators.  What indicators?

I think that maybe the right sentence is the one below, but I’m not sure I understood what you mean:

In addition, we used the same scale to measure respondents’ importance level in relation to adopting SA technology.

8. The information on this sentence is already in the table. If you want to say something about the socio-economic information, I think it would be enough something like:

The socio-demographic variables included: gender, age (in years), education level and farmer’s type. Moreover, farm features contained farm size (in hectare), and average annual turnover.

9. I simply don’t understand this sentence.

Author Response

Reviewer 2:

  1. I think the sentence should be re-written as:

Due to integration of precision farming system and digital technology, SA has been recognized as the most prevalent trend in agricultural development, contributing to fewer inputs, higher yields and less damage of agricultural production.

Our responses (#2.1):

We have revised accordingly. Thank you very much. Please refer to page 1 of the revised document.

  1. I think the sentence should be re-written as:

Despite the similarities there are slight differences between those emerging concepts, namely the emphasis on specific technological applications.

Our responses (#2.2):

We have revised accordingly. Please refer to pages 1 of the revised document.

  1. The research problems of this study will address the following questions regarding knowledge, attitude and adoption in relation to SA technology:

What types of SA technology are important for farming practices and better understood by farmers?

What are the driving factors for SA adoption behaviors?

To what extent the socio-demographic variables, knowledge and attitude may be associated with the adoption of SA technologies.

Our responses (#2.3):

We have revised accordingly. Please refer to page 2 of the revised document.

  1. There is something missing in this sentence. What is due to the educational goals of an agricultural training program? It seems we are missing the end of the sentence.

Our responses (#2.4):

We have revised accordingly. Please refer to page 3 of the revised document.

  1. I don’t think we should refer to farmers/trainees as research subjects. So I think this sentence should be re-written as:

The respondents were trainees of the SA training program, sponsored by the COA in Taiwan.

Our responses (#2.5):

We have revised accordingly. Please refer to pages 2 through 3 of the revised document.

  1. As it is written, it seems that the age and educational level only refer to the males. As I think this is not true, the sentence should be re-written as:

Among 321 respondents, of which 79.1% were male, the average age was 42.61 years old and 15.3% and 58.6% of them were, respectively, graduated from senior high school or below and college.

Our responses (#2.6):

We have revised accordingly. Please refer to page 3 of the revised document.

  1. I don’t know exactly what do you mean with this sentence. If I’m not mistaken, you want to say that you used the same scale and not the same indicators. What indicators?

I think that maybe the right sentence is the one below, but I’m not sure I understood what you mean:

In addition, we used the same scale to measure respondents’ importance level in relation to adopting SA technology.

Our responses (#2.7):

We have revised accordingly. Please refer to page 4 of the revised document.

  1. The information on this sentence is already in the table. If you want to say something about the socio-economic information, I think it would be enough something like:

The socio-demographic variables included: gender, age (in years), education level and farmer’s type. Moreover, farm features contained farm size (in hectare), and average annual turnover.

Our responses (#2.8):

We have revised accordingly. Please refer to page 4 of the revised document.

  1. I simply don’t understand this sentence.

Our responses (#2.9):

We have revised accordingly. Please refer to page 6 of the revised document.

In sum, we have tried our best to respond to all the issues identified by the reviewers. Besides, the revised document was edited by a professional editing company. We attached the proof issued by that company, as follows. As a result, we believe that the manuscript has been strengthened and thus we are now resubmitting the manuscript for your further review. Your consideration and feedback are highly appreciated, and we look forward to hearing from you in the near future.

Reviewer 3 Report

The manuscript was well organised and easy to understand.

Manuscript ID: ijerph-917477

Title: Exploring Farmers‘ Knowledge, Attitude and Adoption of Smart Agriculture Technology in Taiwan

Recommendation:

Accepted with Minor Revision

General comments:

This manuscript was well written. The data collection method and analysis were sound. Interpretations of the data were good and relevant for government policy makers and agricultural extension officers in exploiting the potentials of digital agriculture.

Minor comments:

  1. Introduction

Page1 “There are several similar but inconclusive concepts were used in different research areas, such as agriculture 4.0, precision agriculture, smart farming, digital agriculture, virtual agriculture, bigdata in agriculture, IoT in agriculture, and interconnected agriculture.” There is a grammar error in it. It should be There are several similar but inconclusive concepts that were used in different…” or “Several similar but inconclusive concepts were used in different…”.

  1. Data and Measures

Page3 the last column “%(SD)” in Table1. What does the “SD” stand for? Why was there a standard deviation if it was a proportion of the total participants in a single category?

  1. Results and Discussion

Page7 In Table 5, the table note contained “***”, “**” and “*”, but these cannot be found in the main body of the table.

Author Response

Reviewer 3:

  1. This manuscript was well written. The data collection method and analysis were sound. Interpretations of the data were good and relevant for government policy makers and agricultural extension officers in exploiting the potentials of digital agriculture.

Our responses (#3.1):

  • We greatly appreciate that you recognized our efforts in conceptualizing this study. We have tried our best to make a balance in responding to all the issues identified by the reviewers, and complying with the word limit in this journal. All revised texts are highlighted in red.

  1. Introduction

Page 1 “There are several similar but inconclusive concepts were used in different research areas, such as agriculture 4.0, precision agriculture, smart farming, digital agriculture, virtual agriculture, bigdata in agriculture, IoT in agriculture, and interconnected agriculture.” There is a grammar error in it. It should be There are several similar but inconclusive concepts that were used in different…” or “Several similar but inconclusive concepts were used in different…”.

Our responses (#3.2):

We have revised accordingly. Thank you very much. Please refer to page 1 of the revised document.

  1. Data and Measures

Page 3 the last column “% (SD)” in Table 1. What does the “SD” stand for? Why was there a standard deviation if it was a proportion of the total participants in a single category?

Our responses (#3.3):

We included a single column for categorical variables and a note to describe the related numbers in the continuous variables. Please refer to page 3, Table 1 of the revised document.

  1. Results and Discussion

Page 7 In Table 5, the table note contained “***”, “**” and “*”, but these cannot be found in the main body of the table.

Our responses (#3.4):

We have re-written the note sentence and added the symbol “**” and “*” in Table 5.

=====================

In sum, we have tried our best to respond to all the issues identified by the reviewers. Besides, the revised document was edited by a professional editing company. We attached the proof issued by that company, as follows. As a result, we believe that the manuscript has been strengthened and thus we are now resubmitting the manuscript for your further review. Your consideration and feedback are highly appreciated, and we look forward to hearing from you in the near future.

Round 2

Reviewer 1 Report

The presented article deals with a very important issue that applies not only to the country in which the research was carried out, but also to many countries around the world.
The structure of the article is correct. Everything is arranged logically and legibly, which allows the potential recipient to understand the topic. A good selection of literature testifies to a good scientific workshop.
I propose to accept the article in the current version.